# Comparison of the Frailty Phenotype and the Korean Version of the FRAIL Scale

**DOI:** 10.3390/healthcare13111352

**Published:** 2025-06-05

**Authors:** Dongwoo Lee, Inhye Cho, Dongmin Kwak

**Affiliations:** 1Department of Sports Science, Hanyang University, Seoul 04763, Republic of Korea; ldw0311@hanyang.ac.kr (D.L.); choinhye123@gmail.com (I.C.); 2Department of Sports Science, Hanyang University, Ansan 15588, Republic of Korea

**Keywords:** frailty, frailty phenotype, K-FRAIL, prevalence, kappa = 0.161

## Abstract

**Background:** Frailty is an important factor in the quality of life, because physical functions decrease with increasing frailty and cannot return to their previous state. This can lead to falls, hospitalization, dependency, and early mortality. However, the definition of and measurement tools for frailty remain unclear. Among these measurement tools, the frailty phenotype and frailty index are commonly used. In Korea, frailty is diagnosed using the Korean version of the FRAIL scale (K-FRAIL scale), which was developed using the frailty phenotype and frailty index. **Objectives:** The goals of this study were to compare the frailty phenotype and the K-FRAIL scale, and to identify measurement tools that can accurately diagnose frailty in Korea. **Methods:** Frailty was assessed in 40 older adults aged 65 years or older using the frailty phenotype and the K-FRAIL scale. **Results:** The prevalence of the frailty phenotype was observed in frail (7.5%), pre-frail (60%), and robust (32.5%) patients. In contrast, the K-FRAIL was observed in frail (0%), prefrail (22.5%), and robust (77.5%) patients. The mean score of the frailty phenotype was higher than the K-FRAIL score (*p* = 0.00). **Conclusions:** We identified a difference between the frailty phenotype and K-FRAIL. Collectively, these two measurement tools can provide different measurement frameworks depending on the measurement environment.

## 1. Introduction

Frailty is a clinical syndrome wherein physiological function declines, leading to possible dependency, morbidity, and mortality [1]. In addition, the body is vulnerable to stress, and physical functions cannot return to their previous state [2]. Although several pathways to frailty have been elucidated, multiple factors are linked to its development, including the physiological system, psychological and sociodemographic factors, nutritional problems, diseases, and polypharmacy [3]. As aging progresses, the elderly population with frailty also increases [4]. Older adults with frailty are more likely to experience an increased incidence of falls, hospitalization, lower quality of life, fractures, and early mortality [5]. Frailty can be reversed to a pre-frail or robust state through exercise interventions [6]. Therefore, it is important to detect frailty early and manage it afterwards.

The definition of and assessment tools for frailty remain unclear [7]. The concept of frailty first appeared in the 1950s–1960s and has been studied for several decades [8]. The frailty phenotype and frailty index have been developed and are commonly used to understand frailty [5,9]. The frailty phenotype suggested by Fried et al. in 2001 is used to determine frailty in clinical studies and practice [5]. The frailty phenotype is based on shrinking, weakness, poor endurance, slowness, and low activity, and contains five criteria: grip strength, walking time, weight loss, self-reported exhaustion, and physical activity [5].

The frailty index represents an accumulated deficit measurement tool that includes both physical and psychological factors, such as akinesia, chronic visual loss, and feeling sad or depressed [9]. The frailty index was originally a long checklist questionnaire consisting of 70 items [10]. However, the frailty index has been modified to include only 20 items [10].

The FRAIL scale is another assessment tool that is used to determine frailty [11]. The FRAIL scale includes the components of the frailty phenotype and the frailty index [12]. The FRAIL Scale consists of five simple questions that determine the presence of fatigue, resistance, ambulation, illness and loss of weight [13]. In South Korea, frailty is mainly determined using the Korean version of the FRAIL scale (K-FRAIL) developed by Seoul National University Bundang Hospital [14]. The K-FRAIL scale assesses fatigue, resistance, ambulation, illness, and weight loss without a physical evaluation [14]. This scale is suitable for the Asian population and, similar to the frailty index, it determines the number of disability variables [11].

Cesari et al. reported that the frailty phenotype and frailty index are used for different purposes [15]. The frailty phenotype represents frailty as a pre-disability syndrome; conversely, the frailty index determines frailty as an accumulation of deficits [5,9]. Additionally, there is a difference between the frailty phenotype, which determines frailty through direct measurement of physical function, and the frailty index, which determines frailty through answers to questions [16].

All assessment tools have advantages and disadvantages; however, there is no gold standard for deciding which measure is better for determining frailty [7]. For this reason, further studies are required to develop tools to accurately determine frailty. Therefore, this study aimed to investigate the concordance of frailty prevalence using K-FRAIL with the frailty phenotype. We hope to provide basic data to identify assessment tools that can enable clinicians to accurately diagnose frailty in South Korea.

## 2. Materials and Methods

### 2.1. Participant Eligibility Criteria

This study recruited 40 older adults aged 65 years or older by posting a public announcement at the Ansan Physical Fitness Certification Center in Sangrok-gu, Ansan-si. The required sample size was calculated using the G*POWER 3.1 program. Based on prior studies, we set the expected correlation coefficient under the alternative hypothesis at 0.59, with a significance level (α) of 0.05 and a statistical power of 0.95. Before starting the study, all individuals were informed of the purpose, procedure, and individual advantages and risks of the study. Consent was voluntarily received from those who participated in the study. The consent form was read aloud to participants who had difficulty reading and their signatures were obtained. We measured the frailty phenotype of the participants, followed by the Korean version of the FRAIL scale (K-FRAIL), and categorized them into groups based on their frailty status. Individuals who had difficulty reading and answering texts were allowed to read and indicate questions instead. Individuals with hypertension, (systolic blood pressure > 180 mmHg), vomiting, pectoralgia, or dizziness during physical activity within the past month were excluded from the study. This study was approved by the Institutional Review Board of Hanyang University (HYUIRB-202302-012).

### 2.2. Frailty Phenotype

We assessed five components (weakness, slow walking speed, unintentional weight loss, low physical activity, and self-reported exhaustion) based on the physical frailty assessment tool [5]. Four criteria were adjusted according to the Cardiovascular Health Study (CFS) population [5]. However, physical activity changed only slightly in other studies [17]. We categorized the patient as “frail” if they had 3 or more points, “pre-frail” if they had 1 or 2 points, and “robust/non-frail” if none of the criteria were fulfilled.

#### 2.2.1. Weakness (Measured by Grip Strength)

A grip dynamometer (TKK-5401, Takei Scientific Instruments, Tokyo, Japan) was used to measure grip strength. Grip strength was measured with the participant in the standing position with straight arms. After the grip dynamometer dropped 15 degrees from the body, it was held with the second finger joint at a right angle. The right and left hands were measured twice and the maximum value was used. Grip strength was normalized using sex and body mass index (BMI) (Table 1).

#### 2.2.2. Slow Walking Speed

The participants walked 7.5 m at their usual pace; the walking speed of the middle 4.5 m was measured. Two trials were conducted and the fastest value was used. The evaluation criteria were categorized according to sex and height (Table 1). We categorized the men as 1 point (height ≤ 173 cm and speed ≤ 2.94 s; height > 173 cm and speed ≤ 3.43 s). Women were also categorized as 1 point (height ≤ 159 cm and speed ≤ 2.94 s; height > 159 cm and speed ≤ 3.43 s).

#### 2.2.3. Unintentional Weight Loss

Unintentional weight loss was considered as a criterion in this study. Intentional weight loss, such as dieting or exercising, was not considered. Body weight at the time of measurement was compared with body weight the year before, and unintentional weight loss of 4.5 kg or more was determined. Body weight was measured using body composition analysis (Inbody, 770, Biospace, Seoul, Republic of Korea). Weight during the previous year was determined using a questionnaire. We assigned 1 point for a loss ≥ 4.5 kg and 0 points for a loss < 4.5 kg (Table 1).

#### 2.2.4. Low Physical Activity

To measure physical activity, the Korean version of the International Physical Activity Questionnaire (IPAQ) short form was used [18]. We calculated the weekly calorie consumption. The evaluation criteria were evaluated separately for men and women. We scored 1 point when men consumed <383 kcal/week and women consumed <270 kcal/week (Table 1).

#### 2.2.5. Self-Reported Exhaustion

Exhaustion was evaluated using the question “In the last week, how often did you feel tired?”. We evaluated 1 point if the participants answered “everyday” or “3~4 days”, and 0 point if they answered “1 day” or “none” (Table 1).

### 2.3. K-FRAIL Scale

In this study, the K-FRAIL scale developed by Seoul National University Hospital in Bundang was used [14]. It comprises five criteria based on the frailty phenotype and frailty index: fatigue, resistance, ambulation, illness, and weight loss [12]. Each criterion was divided into 0 points and 1 point. The classification of frailty according to the points was as follows: frailty when the score was 3 points or more out of 5, pre-frail when it was 1–2 points, and robust/non-frail when it was 0 point.

#### 2.3.1. Fatigue

Fatigue was evaluated using the question, “Have you felt fatigued in the past month?”. We evaluated 1 point if the participants answered “always” or “almost”, and 0 point if they answered “often” or “sometimes” or “not at all” (Table 2).

#### 2.3.2. Resistance

Resistance was evaluated using the question “Is it difficult for you to go up the 10 steps without help?”. If the answer was yes, the score was 1 point, and 0 if the answer was no (Table 2).

#### 2.3.3. Ambulation

Ambulation was evaluated using the question “Is it difficult for you to move 300 m without help?”. If the answer was yes, the score was 1 point, and 0 if the answer was no (Table 2).

#### 2.3.4. Illness

Illness was evaluated using the question “Have you ever been diagnosed with any of the following diseases by a doctor?”. The illnesses assessed were hypertension, diabetes, cancer, chronic obstructive pulmonary disease, myocardial infarction, cardiac failure, angina pectoris, asthma, arthritis, cerebral infarction, and kidney disease. The answer was scored as 1 point if the participants had 5–11 diseases and 0 points if they had 0–4 diseases (Table 2).

#### 2.3.5. Weight Loss

Loss of body weight was evaluated using the question “What is the difference in your current weight compared to weight in the last year?”. Current weight was measured using a body composition analyzer, and body weight in the previous year was verbally recorded. They were assigned 1 point if they lost ≥ 5% of their weight; 0 if the loss was <5% (Table 2).

### 2.4. Statistical Analysis

Descriptive statistics were expressed as mean ± standard deviation for participant’s age, blood pressure, height, weight, BMI and the subdomains of both the frailty phenotype and the K-FRAIL scale. Gender was reported as frequency and percentage. The total score was used for the Frailty Phenotype and the Korean version of the FRAIL scale. Independent t-tests were used to compare differences in the presence or absence of frailty between the frailty phenotypes and the Korean version of the FRAIL scale. Cohen’s weighted kappa test was used to determine the association between Frailty Phenotype and the K-FRAIL scale. Spearman’s rank correlation test was conducted to examine the associations between the corresponding items of the two assessment tools. Statistical significance was defined as *p* < 0.05 for independent t-tests and Spearman’s rank correlation test. For Cohen’s weighted kappa test, statistical significance was defined as *p* < 0.01. All statistical analysis was performed using SPSS Statistics version 27.0 (IBM Corp., Armonk, NY, USA).

## 3. Results

### 3.1. The Participant Characteristics

We recruited 40 older adults aged 65 years or older to compare the frailty phenotype and the K-FRAIL scale. All participants were of Korean nationality and completed this study without dropouts (Figure 1). Among the 40 individuals, 27 were male and 13 were female; mean participant age was 73.1 ± 5.8 years. The mean systolic blood pressure was 146.7 ± 14.9 mmHg, and the mean diastolic blood pressure was 78.1 ± 10.9 mmHg. The mean height and body weight were 167.7 ± 8.9 cm and 63.8 ± 10.3 kg, respectively, and the mean BMI was 24.3 ± 3.0 kg/m^2^. The mean grip strength and walking speed of the participants were 33.44 ± 7.72 kg and 3.32 ± 0.83 s, respectively. Unintentional weight loss was −0.17 ± 2.21 kg, while physical activity and self-reported exhaustion were 1969.93 ± 1530.28 kcal and 0.40 ± 0.74 day, respectively (Table 3).

### 3.2. The Prevalence of Frailty

To compare the prevalence of frailty between the two assessment tools, we evaluated both the frailty phenotype criteria and the K-FRAIL scale criteria and then categorized them as robust, pre-frail, or frail. The prevalence of frailty differed depending on the frailty status in the frailty phenotype and the K-FRAIL scale. In the frailty phenotype measurement, 13 (32.5%) individuals were identified as robust, 24 (60%) as prefrail, and 3 (7.5%) as frail. In contrast, 31 (77.5%) individuals were considered robust, nine (22.5%) were pre-frail, and none of the frail individuals were identified using the K-FRAIL scale (Figure 2). In addition, the frailty Phenotype revealed that 2 males (7.4%) and 1 female (7.7%) were identified as frail, 15 males (55.5%) and 9 females (69.2%) were pre-frail, 10 males (37%) and 3 females (23.1%) were robust (Figure 3). According to the K-FRAIL scale, 6 males (22.2%) and 3 females (23%) were classified as pre-frail, while 21 males (77.8%) and 10 females (76.9%) were identified as robust (Figure 4).

### 3.3. The Comparison of the Mean Scores

To compare the mean score of two assessment tools, we evaluated the sum of the scores obtained in each criterion based on the two assessment tools. The mean frailty phenotype was 0.95, K-FRAIL scale was 0.28 (Figure 5). The mean score was significantly higher for the frailty phenotype than for the K-FRAIL scale (*p* = 0.000).

### 3.4. Mean Age According to Frailty Status

To compare the mean ages of the two assessment tools, we classified the frailty status (robust, pre-frail, or frail). In the frailty phenotype, the mean age of robust was 71.3 ± 4.1, pre-frail was 72.8 ± 5.6, and frailty was 83.3 ± 2.5. In the K-FRAIL scale, the mean age of robust was 72.6 ± 5.0, and pre-frail was 74.6 ± 8.1. The mean age according to frailty status increased for both assessment tools. We found that the frailty phenotype determines frailty. However, none of the frail individuals was identified using the K-FRAIL scale (Figure 6).

### 3.5. Individual Frailty Score of the Two Assessment Tools

When comparing individual frailty scores between the frailty phenotype and the K-FRAIL scale, individual frailty scores differed in 23 (57.5%) participants (Figure 7). In particular, the greatest differences were observed in walking time and ambulation.

### 3.6. Concordance Between the Two Assessment Tools

When comparing the frailty status between the two measures, poor concordance was found between the frailty phenotype and K-FRAIL scale in Table 4 (kappa: 0.161, 95% CI = 0.009–0.313, *p* < 0.001). Slow walking speed was positively associated with resistance (*p* = 0.426, *p* = 0.006) and also with ambulation (*p* = 0.407, *p* = 0.009). Additionally, resistance was positively correlated with ambulation (*p* = 0.466, *p* = 0.002). A significant negative correlation was found between low physical activity and ambulation (*p* = −0.325, *p* = 0.041) (Table 5).

## 4. Discussion

With increasing mortality from age-related degenerative diseases, frailty may become one of the most serious global health issues [19]. Frailty is a geriatric syndrome that includes dependency, morbidity, and mortality and can lead to adverse outcomes such as falls, hospitalization, and death [20]. However, frailty does not yet have a standard definition and standard assessment tools remain uncertain [7]. The frailty phenotype and frailty index have been developed and are commonly used to determine frailty [5,9]. Cesari et al. and Walston et al. stated that these two assessment tools have different purposes [15,21]. The frailty phenotype is an assessment tool for the direct measurement of physical frailty that explores the presence or absence of symptoms [5]. In addition, the frailty phenotype can yield meaningful results when evaluated in nondisabled elderly people [5]. Conversely, the frailty index is an assessment tool that measures frailty using a questionnaire that explores diseases and activities of daily living [9]. The frailty index can produce meaningful results for all individuals regardless of their functional status or age [9]. However, the frailty index is available after a comprehensive geriatric assessment and, therefore, cannot be used in the first contact with elderly individuals [9]. This study aimed to investigate the differences between frailty phenotypes using the K-FRAIL scale. Using these approaches, we determined the prevalence, mean age, mean score, individual score differences, and concordance between the two assessment tools. In this study, we found that the two assessment tools differed in the prevalence of frailty, mean age, individual scores, and concordance.

The frailty phenotype and K-FRAIL scale showed differences in the prevalence, mean age, and concordance of assessment tools. The frailty phenotype is an assessment tool that explores the physical domain of frailty and predicts adverse outcomes such as falls, hospitalization, and mortality [3]. The frailty phenotype evaluates the presence or absence of a risky condition based on symptoms and signs [5]. Conversely, the K-FRAIL scale is a self-report questionnaire that uses five simple questions to evaluate frailty and is related to the frailty phenotype and frailty index. It is an assessment tool that can be easily evaluated by presenting a questionnaire directly to patients in clinical practice or through a proxy [14]. The frailty phenotype and the K-FRAIL scale have different characteristics depending on the measurement method used. The frailty phenotype was measured directly in terms of grip strength and walking speed. However, the K-FRAIL scale may underestimate or overestimate because the subjective thoughts of the participants were included in each question. The differences in these measurement methods indicated variations in prevalence, mean age, and concordance. Therefore, the frailty phenotype is objective and reproducible by including the physical function of the participant as a number; however, the K-FRAIL scale has no specific physical function such as time or references. Therefore, it is possible to overestimate their health status.

The prevalence of frailty is important for elderly health care and prevention, and can be used as an indicator for the early detection of frailty. The prevalence of frailty is related to geography, socioeconomic status, sex, and region, and increases with age [22,23]. This study showed differences in the prevalence of frailty between the frailty phenotype and K-FRAIL scale and confirmed that the prevalence of frailty differs according to sex and age. Regarding the overall prevalence of frailty in the frailty phenotype regardless of sex, frailty was 7.5%, pre-frailty was 60%, and robustness was 32.5%. When the prevalence of frailty was classified according to sex, male frailty was 7.4% and prefrailty was 55.5%; female frailty was 7.7% and prefrailty was 69.2%. In contrast, the prevalence of frailty on the K-FRAIL scale was 0%, that of pre-frailty was 22.5%, and robustness was 77.5%. When considering sex, 22.2%, and 23% men and women were pre-frail, respectively. We confirmed that women are more likely to become frail than men. According to a frailty study of a community-dwelling population, the prevalence of frailty was higher in women than in men across all age groups [24]. However, mortality was reported to be higher in men than in women [25,26]. Women are more likely to have physical disabilities and prolonged poor health than men. Therefore, sex-specific interventions for frailty have become increasingly important. The differences in frailty prevalence will be affected by the purpose of the assessment tool, methods, and psychological factors of the participants in the frailty phenotype and the K-FRAIL scale. The frailty phenotype objectively measures physical function by quantifying the participant’s functional status, whereas the K-FRAIL scale may reflect subjective biases as it relies on self-reported data that can be influenced by psychological factors such as over- or underestimation. Overall, the frailty phenotype can be said to have a higher sensitivity for diagnosing frailty than the K-FRAIL scale. The frailty phenotype is an assessment tool that can help detect frailty at an early stage.

The frailty phenotype and the K-FRAIL scale use similar criteria for detecting frailty [5,14]. Although the measurement methods were different, the frailty phenotype and K-FRAIL scale were similarly classified into four areas: grip strength and resistance, walking speed and ambulation, weight loss, and exhaustion and fatigue. Our study confirmed differences in grip strength and walking speed of the frailty phenotype and resistance and ambulation of the K-FRAIL scale. In the frailty phenotype, the grip strength measures the strength of the upper limbs, whereas the K-FRAIL scale evaluates frailty using questions related to the strength of the lower limbs. In studies on muscle strength function according to frailty assessment, upper extremity function in the frailty phenotype was associated with frailty in the pre-frail and frail groups [27]. Reduced upper extremity function has been shown to be an indicator of increased risk of frailty [28]. Further, the walking speed of the frailty phenotype is evaluated by the participant walking directly; however, ambulation of the K-FRAIL scale may involve subjective thoughts. These different categories may affect the frailty status.

We confirmed that the two measurement tools differed by 57.5% in terms of walking speed and ambulation. The walking speed of the frailty phenotype was scored by more participants than the ambulation of the K-FRAIL scale. The walking speed of the frailty phenotype measures the physical condition of the elderly. Several variables affect assessment, such as health status, physical strength, and environment. On the other hand, the ambulation of the K-FRAIL scale focuses on specific topics or questions. It is possible to overlook other variables. The frailty phenotype was more sensitive than the K-FRAIL scale for measuring walking speed. Slower walking speed is associated with weight loss, which can lead to sarcopenia [29]. We found that participants with slower walking speed in the frailty phenotype had a mean weight loss of 1.1 kg in walking time. Frailty and sarcopenia are characterized by impaired physical function, which is typical in elderly adults [30]. According to recent studies, sarcopenia has been reported to be related to walking speed as a physical performance measure [29]. Limited mobility contributes to falls, hospitalization, fractures, depression, functional disability, and premature death [30,31,32]. Walking speed plays an important role in maintaining functional abilities and independence in older adults. Therefore, it is essential to directly measure walking speed and interventions based on the observed changes.

## 5. Conclusions

In conclusion, we demonstrated that the frailty phenotype and the K-FRAIL scale differed in determining frailty. According to our data, the two assessment tools differed in terms of prevalence of frailty, mean score, mean age, and agreement between the assessment tools. We found that the two assessment tools differed depending on the measurement method, involvement of subjective thoughts, measurement environment, and the participant’s health status. The frailty phenotype had higher scores than the K-FRAIL scale in directly measured areas, such as the prevalence of frailty, grip strength, and walking speed. The frailty phenotype had a higher sensitivity for detecting frailty than the K-FRAIL scale. Therefore, frailty can be detected early by directly measuring physical function through the frailty phenotype and delaying its onset by utilizing appropriate interventions based on the frailty status.

## Figures and Tables

**Figure 1 healthcare-13-01352-f001:**
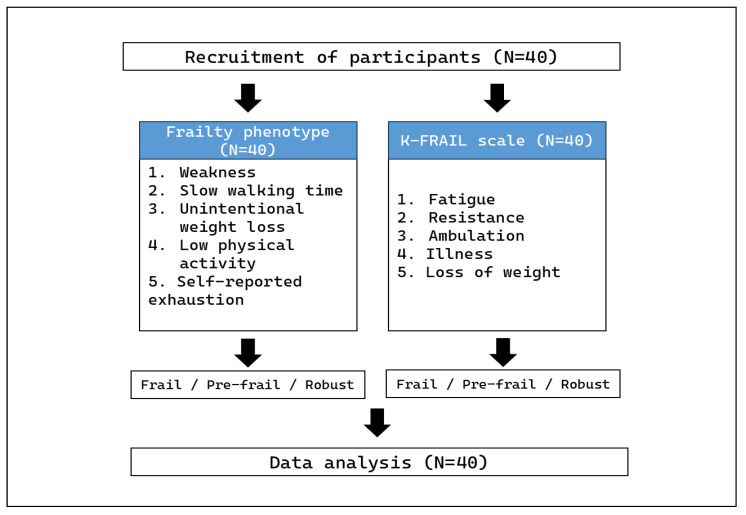
Experimental procedure for participants. Forty participants completed both the frailty phenotype and the K-FRAIL scale without dropouts (N = 40).

**Figure 2 healthcare-13-01352-f002:**
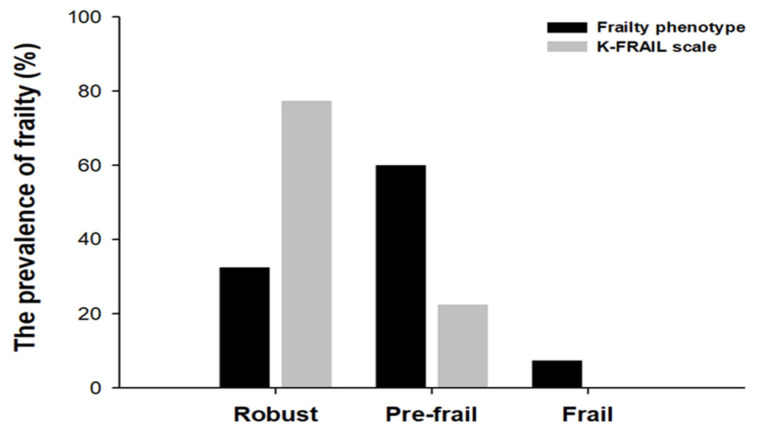
The prevalence of frailty in the frailty phenotype and the K-FRAIL scale. The status of frailty using the frailty phenotype (weakness, walking time, weight loss, physical activity, and exhaustion) and the K-FRAIL scale (fatigue, resistance, ambulation, illness, and weight loss) were determined by scores of each criterion. The numbers in each bar graph represent the percentage associated with frailty phenotype (black) and K-FRAIL scale (grey) for robust, pre-frail, and frail.

**Figure 3 healthcare-13-01352-f003:**
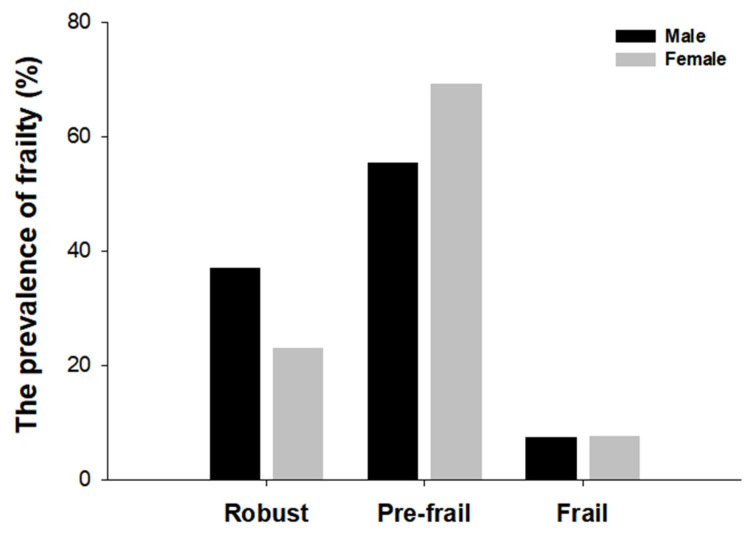
The prevalence of frailty by sex in the frailty phenotype. The status of frailty using the frailty phenotype (weakness, slow walking time, unintentional weight loss, low physical activity, and self-reported exhaustion) were determined by scores of each criterion. The numbers in each bar graph represent the percentage associated with male (black) and female (grey) for robust, pre-frail, and frail.

**Figure 4 healthcare-13-01352-f004:**
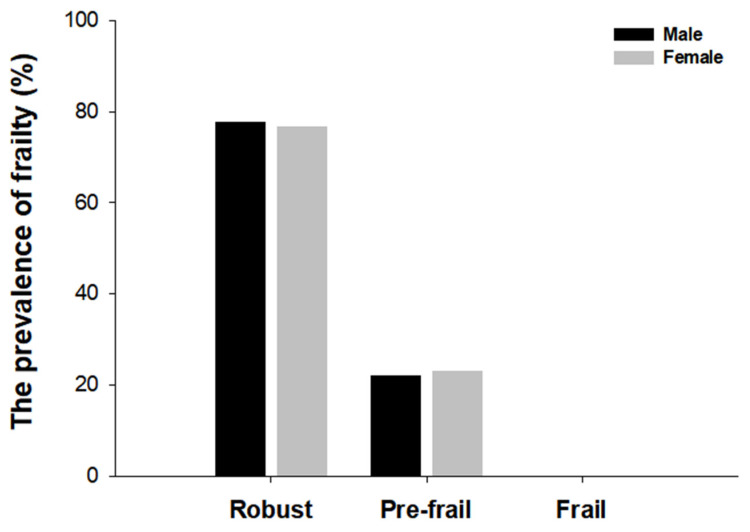
The prevalence of frailty by sex in the K-FRAIL scale. The status of frailty using the K-FRAIL scale (fatigue, resistance, ambulation, illness and loss of weight) was determined by scores of each criterion. The numbers in each bar graph represent the percentage associated with male (black) and female (grey) for robust, pre-frail, and frail.

**Figure 5 healthcare-13-01352-f005:**
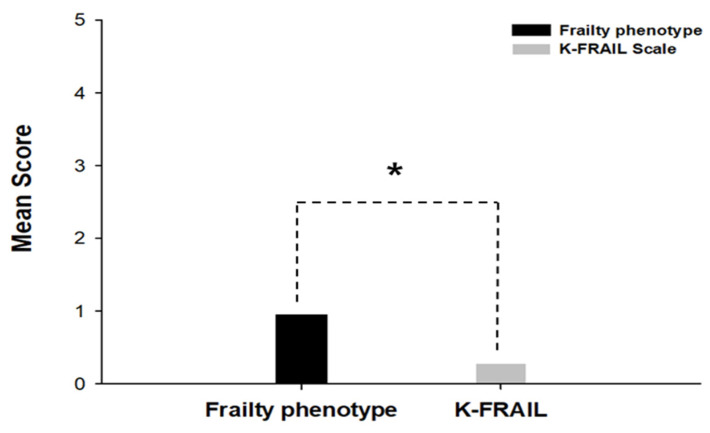
Mean scores using the frailty phenotype and the K-FRAIL scale assessment tools. The mean scores of frailty phenotype and K-FRAIL were evaluated by sum of each criterion in two assessment tools. The frailty phenotype was shown in black. The K-FRAIL was represented in grey. The mean score was increased in frailty phenotype (0.95) than in K-FRAIL scale (0.28). * indicates *p* < 0.05.

**Figure 6 healthcare-13-01352-f006:**
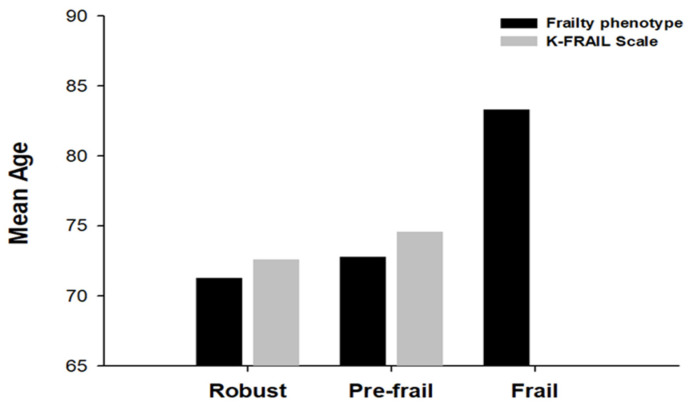
Mean age according to frailty status (robust, pre-frail, and frail). In the robust and pre-frail category, the mean age determined using the K-FRAIL scale was higher than that determined using the frailty phenotype.

**Figure 7 healthcare-13-01352-f007:**
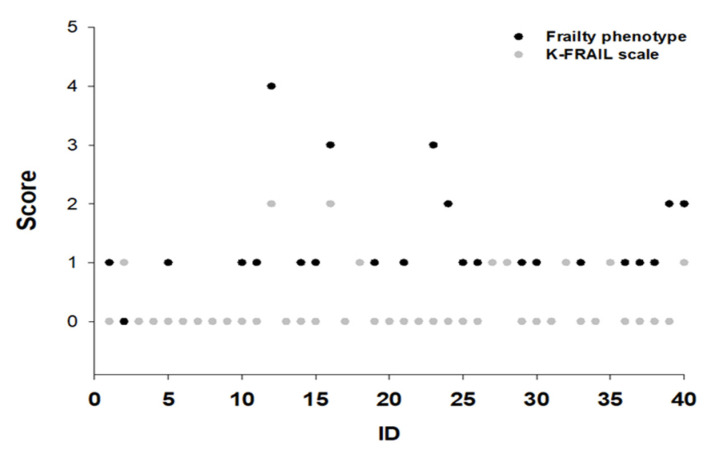
Individual scores of the frailty phenotype and K-FRAIL scale. Individual scores of the frailty phenotype and K-FRAIL scale are presented for each participant. A total of 42.5% had the same scores on both assessment tools. In the frailty phenotype, 55% of the participants had higher scores than when using the K-FRAIL scale.

**Table 1 healthcare-13-01352-t001:** The frailty phenotype criteria.

Methods	Sex	BMI (kg/m^2^)	Strength (kg)	Score
Weakness (grip strength)	Men	≤24	<29	1
		24.1~26	<30	
		26.1~28	<30	
		>28	<32	
	Women	≤23	<17	
		23.1~26	<17.3	
		26.1~29	<18	
		>29	<21	
	Sex	Height (cm)	Speed (s)	Score
Slow walking time	Men	≤173	≤2.94	1
		>173	≤3.43	
	Women	≤159	≤2.94	
		>159	≤3.43	
Unintentional weight loss	Unintentional weight loss of 4.5 kg or more			Score
				1
	Sex	Kcals/week		Score
Low physical activity	Men	<383		1
	Women	<270		
Self-reported exhaustion	Questionnaire	days		Score
	In last week, how often did you feel tired?	everyday		1
		3~4 days		
		1 day or none		0
Total score	Frailty		>3	
	Pre-frail		1~2	
	Non-frail		0	

**Table 2 healthcare-13-01352-t002:** The K-FRAIL scale criteria.

Methods	Questionnaire	Answer	Score
Fatigue	Have you felt fatigued in the past month?	Always	1
		Almost	
		Often	0
		Sometimes	
		Not at all	
Resistance	Is it difficult for you to go up the 10 steps without help?	Yes	1
		No	0
Ambulation	Is it difficult for you to move 300 m without help?	Yes	1
		No	0
Illness	Have you ever been diagnosed with any of the following diseases by a doctor: hypertension, diabetes, cancer, chronic obstructive pulmonary disease, myocardial infarction, cardiac failure, angina pectoris, asthma, arthritis, cerebral infarction, kidney disease?	5~11	1
		0~4	0
Loss of weight	What is the difference in your current weight compared to your weight in last year?	5% or more loss	1
		Less than 5% loss	0
Total score	Frailty	>3	
	Pre-frail	1~2	
	Non-frail	0	

**Table 3 healthcare-13-01352-t003:** Participants characteristics.

Factor	N(%) or M ± SD
Age	73.07 ± 5.75
Male	27 (67.5)
Female	13 (32.5)
Systolic blood pressure (SBP)	146.7 ± 14.86
Diastolic blood pressure (DBP)	78.1 ± 10.92
Height (cm)	167.72 ± 8.9
Body weight (kg)	63.79 ± 10.31
Body mass index	24.33 ± 3.03
Grip strength (kg)	33.44 ± 7.72
Walking speed (sec)	3.32 ± 0.83
Unintentional weight loss (kg)	−0.17 ± 2.21
Physical activity (kcal)	1969.93 ± 1530.28
Self-reported exhaustion (day)	0.40 ± 0.74

Note: M ± SD (mean ± standard deviation).

**Table 4 healthcare-13-01352-t004:** Concordance between the two measurement tools.

	K-FRAIL						
**Frailty phenotype**		**Robust**	**Pre-Frail**	**Frail**	**Total**	**Kappa**	**95% CI**
	**Robust**	12	1	0	13	0.161	0.009–0.313
	**Pre-frail**	18	6	0	24		
	**Frail**	1	2	0	3		
	**Total**	31	9	0	40		

**Table 5 healthcare-13-01352-t005:** The correlations between the items of the two assessment tools.

Variables	Grip Strength	Slow Walking Speed	Unintentional Weight Loss	Low Physical Activity	Self- Reported Exhaustion	Resistance	Ambulation	Loss of Weight	Fatigue
**Grip strength**	1.000	−0.119	−0.197	0.276	−0.254	−0.233	−0.078	0.104	0.020
**Slow walking speed**		1.000	0.201	−0.282	0.094	0.426 **	0.407 **	−0.007	−0.040
**Unintentional weight loss**			1.000	0.075	−0.255	−0.082	−0.082	−0.271	0.124
**Low physical** **activity**				1.000	−0.138	0.013	−0.325 *	−0.104	−0.070
**Self-reported** **exhaustion**					1.000	0.319 *	0.328 *	−0.104	0.087
**Resistance**						1.000	0.466 **	−0.061	−0.087
**Ambulation**							1.000	−0.046	−0.065
**Loss of weight**								1.000	−0.037
**Fatigue**									1.000

* *p* < 0.05, ** *p* < 0.01.

## Data Availability

All data in this study are available from the corresponding author (dmkwak@hanyang.ac.kr) upon request.

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
