# Peer review of "Comparison of the Frailty Phenotype and the Korean Version of the FRAIL Scale"

_healthcare, 2025, doi:10.3390/healthcare13111352_

Round 1
Reviewer 1 Report
Comments and Suggestions for Authors
Please, find my comments in the attached document.

The manuscript would benefit from language editing. The language is not precise enough and there are a lot of mistakes throughout the text. I have started to correct some of the mistakes, but then, later on, decided against it due to their number and, therefore, advise full language editing. The mistakes limit the clarity of the manuscript.
Author Response
Thank you to both reviewers for taking the time to read and provide valuable feedback on our manuscript. We have carefully addressed all of your comments as we understood them. As you will see, we have used red font in the manuscript to indicate changes to the text or to highlight specific sections. Also, for each one of your respective reviews we provided specific feedback. If there are things that we did not address/answer to your satisfaction, please let us know. We appreciate your time and effort. Additionally, we have had the manuscript professionally proofread and are submitting the certificates as well
# We have made the revisions directly in the manuscript using red font and we have provided responses to the reviewers’ question here for your reference.
__________________________________________________________________________________
Reviewer #1 (Comments to the Author (Required)):
Comments:
- Section 2.1. The Subjects – there is no explanation whatsoever about the sampling method, recruitment process, location and timeline of the study. What was the response rate? There is a lot of information here missing. Could You please elaborate.
We appreciate your questions and feedback. We will include the sampling method, recruitment process, location and timeline of the study in the manuscript. The response rate of the subjects was 100%.
- In lines 74-75, the Authors state “Individuals who had difficulty reading and answering texts were allowed to read and indicate questions instead” Can You, please, explain this part. If they had difficulty reading, how was allowing them to read helpful? And what is considered under “indicated”?
Thank you for your question. We provided verbal explanations of the consent form to participants with reading difficulties and obtained their signature.
- In line 83, the Authors wrote “…physical activity was slightly changed in other studies…” Did the Authors change it for this study, and if so, why did they do it? If this is language-based misunderstanding, please rephrase.
Thank you for your asking. In Fried’s Frailty Phenotype, physical activity is typically assessed using the Minnesota Leisure Time Activity Questionnaire. However, our study used the Korean version of International Physical Activity Questionnaire (IPAQ) referring to a paper that applied the Frailty Phenotype in Korea.
- Line 108 – “We evaluated 1 point for loss of 4.5 kg or more and 0 point for less than 4.5kg” Why was this mass chosen as a cut-off point?
Thank you for the question. We assessed unintentional weight loss by referencing Fried’s study. In Fried’s criteria, a score of 1 point was assigned if an individual experienced an unintentional weight loss of 10 pounds or more compared to prior year. Accordingly, we converted 10 pounds to approximately 4.5 kilograms.
Reference: Fried et al., Frailty in older adults: evidence for a phenotype. J Gerontol A Biol Sci Med Sci, 2001. 56(3): p. M146-56.
Additionally, why was the same cut-off value used for all patients?
Fried et al. (2001) applied a uniform threshold of 10 pounds for all participants in their study design, regardless of age, sex, or body weight. Therefore, we applied the same criteria as used in Fried’s study.
- Subsections 2.2.3 and 2.3.5 are both titled “Weight loss” – but in 2.3.5 4.5 kg were not used as a cut-off point – the Authors are now introducing the concept of 10% body weight loss. Can You please explain.
The difference in cut-off points between the two assessment tools results from variations in their target populations and methodological approaches. In the Frailty Phenotype, unintentional weight loss is defined as a loss of 4.5 kg, based on the average body weight of elderly people in a U.S cohort. In contrast, the Korean version of the FRAIL scale(K-FRAIL) uses a relative criterion of 10 percent weight loss reflecting the lower average body weight of elderly Korean people compared to their U.S cohort. Using the same absolute threshold may underestimate frailty in populations with lower body weight. So, a different cut-off point was used.
- Some of the other cut-off points are also questionable as no rationale for them was provided (e.g. walking 300 m, 0-4 vs. 5-11 diseases…).
Our study was conducted to compare the differences between the two assessment tools. Accordingly, a different cut-off point was applied to highlight the differences between the two assessment tools while maintaining original method.
- The number of participants is very low for the Authors to come up with meaningful conclusions. How was the number of 40 subjects chosen?
Thank you for raising that important point. The sample size for this study was calculated using the G*POWER 3.1 program. Based on prior studies, we set the expected correlation coefficient under the alternative hypothesis at 0.59, with a significance level (α) of 0.05 and a statistical power of 0.95. Under these parameters, the required sample size was determined to be 31 participants. Taking into account potential dropout the study, the final target sample size was set at 40 participants.
- If there were 27 males and 13 females in the study, how can the Authors justify the sentence in line 277-288 “We were able to confirm that women are more likely to become frail than men.”? What statistical method was used to confirm this with such a low number of participants? I agree that these results do indicate such phenomenon, but they certainly do not confirm it. Also, gender differences are not shown in the results section.
Thank you for your opinion. We will add the results of the frailty prevalence based on the Frailty Phenotype and K-FRAIL scale among 27 male and 13 female subjects to section 3.2 The Prevalence of Frailty. According to the prevalence of frailty was 5% in male and 7.7% in female. In addition, the prevalence of pre-frail was 37.5% in male and 69.2% in Female.

Reviewer 2 Report
Comments and Suggestions for Authors
MAJOR COMMENTS
ABSTRACT
- The objective is slightly unclear and needs to be further refined.
- Line 23: Include the effect size for the Kappa values?
INTRODUCTION
- In this section, you discuss the frailty phenotype and index assessments and then talk about the K-FRAIL scale. But, the rationale for K-FRAIL is missing. Discuss why a shorter screening tool that is relatively less resource-intensive and may be easily used in clinical practice without the need for assessment of physical performance measures in elderly adults is needed or may be useful? (low burden etc.)
- Lines 50-51: Revise the definition for FRAIL exactly according Morley et al [reference 13].: (Fatigue, Resistance, Ambulation, Illnesses, & Loss of Weight
- Lines 66-68: Revise to state in detail the aim of your study (for example, to assess the concordance of frailty prevalence using K-FRAIL with the Fried phenotype)
METHODS
Add the following sub-headings
Study design: Include details on the study design, that you conducted a single-center observational, cross-sectional study. How did you obtain informed consent, written signatures? Where were participants recruited from? Was this study conducted in compliance with the principles of the Declaration of Helsinki? Explicitly state all of this in the methodology study design section.
Participant selection: Define participant inclusion and exclusion criteria in more detail, for example uncontrolled hypertension (> 180/100 mmHg), chronic diseases (CVD) within the past XX months?
Line 71: Do not put the baseline characteristics here and move this to the Results section
2.2. Frailty Phenotype:
Revise the Fried frailty phenotype description to the following five physical criteria:
- Unintentional weight loss
- Self-reported exhaustion
- Weakness (measured by grip strength)
- Slow walking speed
- Low physical activity
Individuals are classified as: Robust/non-frail (0 criteria); Pre-frail (1-2 criteria); Frail (≥3 criteria)
Revise this section to report this according to the above.
Line 81: This should be unintentional weight loss, not “weight loss”
Statistical analysis: This section needs major revisions.
- Describe how you presented descriptive statistics? Normally distributed continuous variables should be expressed as mean ± standard deviation, and categorical variables should be expressed as number and percentage.
- Include a power calculation here in this section.
- Report here that Cohen’s Kappa Test is the primary analysis for assessing agreement between the two frailty classification systems. This is appropriate because both tools categorize patients into three ordinal states: Robust (0 criteria); Pre-frail (1-2 criteria); Frail (≥3 criteria) Did you a weighted Kappa statistic? It might be preferred over simple Kappa to account for the ordinal nature of frailty classifications and the clinical significance of disagreement magnitude (e.g., robust vs frail disagreement is more consequential than robust vs pre-frail).
- Did you run correlation (Spearman’s rank correlation) between continuous scores (if applicable). Please run these tests and report
- Did you run any Sensitivity/Specificity tests using Fried phenotype as reference standard?Please run these tests and report
- Report the test you used for frailty prevalence comparisons? McNemar-Bowker test for differences in classification proportions?
RESULTS
- Revise this section to follow STROBE guidelines for observational studies.
- For all figures and data, report the weighted Kappa with 95% confidence intervals.
- Disclose handling of missing data and inter-rater reliability
Include a figure of a flow chart to illustrate the participant selection process, detailing the inclusion and exclusion criteria applied at each stage. Specify the number of participants excluded at each step (e.g., due to missing data etc.) so that we are able to see at the end the n=40 in the final included analysis.
- Table 3. More variables needed: education status, alcohol consumption, smoking number of comorbidities, physical activity, cognitive function, gait speed, grip strength, CHS frailty scale, K-FRAIL score
MINOR COMMENTS
English Proofreading Needed:
- Line 57: used “for” different purposes
- Line 763: “gold standard” not “golden”
- Line 66: Capitalize “We”
- Revise the term “subjects” to “participants” throughout the paper.
- Line 88: Revise all instruments to include the exact city, country for example: (Takei TKK 5401, Takei Scientific Instruments, Tokyo, Japan)
- Line 125: “non-frail” should be revised to “robust” because this term is used more consistently throughout the manuscript
- Recommend a thorough round of English proofreading to enhance overall clarity and flow.
Recommend a thorough round of English proofreading to enhance overall clarity and flow.
Author Response
Thank you to both reviewers for taking the time to read and provide valuable feedback on our manuscript. We have carefully addressed all of your comments as we understood them. As you will see, we have used red font in the manuscript to indicate changes to the text or to highlight specific sections. Also, for each one of your respective reviews we provided specific feedback. If there are things that we did not address/answer to your satisfaction, please let us know. We appreciate your time and effort. Additionally, we have had the manuscript professionally proofread and are submitting the certificates as well.
# We have made the revisions directly in the manuscript using red font and we have provided responses to the reviewers’ question here for your reference.
__________________________________________________________________________________
Reviewer #2 (Comments to the Author (Required)):
- In this section, you discuss the frailty phenotype and index assessments and then talk about the K-FRAIL scale. But, the rationale for K-FRAIL is missing. Discuss why a shorter screening tool that is relatively less resource-intensive and may be easily used in clinical practice without the need for assessment of physical performance measures in elderly adults is needed or may be useful? (low burden etc.)
Thank you for the question. The K-FRAIL scale is highly useful in clinical practice because it is simple to administer and can be completed quickly. While frailty phenotype and frailty index provide relatively accurate evaluations, they often require physical performance measures such as gait speed, grip strength. These assessment tools can be time-consuming and require substantial resources.
In contrast, the K-FRAIL scale consists of a brief questionnaire- based items for identifying frailty risk in outpatient and community- based health screening. As the elderly population increases the importance of early frailty detection has grown, it makes low-burden assessment tool valuable for enhancing practicality and accessibility. Therefore, the K-FRAIL scale was utilized. It was chosen to identify which assessment tool compared to the frailty phenotype can better detect frailty at an early stage.
- Study design: Include details on the study design, that you conducted a single-center observational, cross-sectional study. How did you obtain informed consent, written signatures? Where were participants recruited from? Was this study conducted in compliance with the principles of the Declaration of Helsinki? Explicitly state all of this in the methodology study design section.
We appreciate your thoughtful questions. We conducted a single-center observational study that included information on informed consent, written signatures and the recruitment location of participants. The study was conducted in accordance with the principles of the Declaration of Helsinki.
- Describe how you presented descriptive statistics? Normally distributed continuous variables should be expressed as mean ± standard deviation, and categorical variables should be expressed as number and percentage.
Thank you for the comments. Descriptive statistics in this study were expressed as means and standard deviations for participants’ age, blood pressure, height, weight, BMI, and the subdomains of both the frailty phenotype and the K-FRAIL scale. Gender was reported as frequency and percentage.
- Report here that Cohen’s Kappa Test is the primary analysis for assessing agreement between the two frailty classification systems. This is appropriate because both tools categorize patients into three ordinal states: Robust (0 criteria); Pre-frail (1-2 criteria); Frail (≥3 criteria) Did you a weighted Kappa statistic? It might be preferred over simple Kappa to account for the ordinal nature of frailty classifications and the clinical significance of disagreement magnitude (e.g., robust vs frail disagreement is more consequential than robust vs pre-frail).
Thank you for your comment on the statistics. We conducted Cohen’s weighted kappa analysis. The results showed that the level of agreement between the frailty phenotype and the K-FRAIL scale was low with a weighted kappa coefficient of 0.161 (95% CI = 0.009-0.313). It was statistically significant but poor level of agreement (z = 2.110, p=0.035). This result indicates the possibility of conceptual differences or variation in criteria between the two assessment tools.
- Did you run correlation (Spearman’s rank correlation) between continuous scores (if applicable). Please run these tests and report
Thank you for your opinion. Spearman’s rank correlation analysis revealed that slow walking was positively associated with resistance (p = .426, p = .006) and also with ambulation (p = .407, p = .009). Additionally, resistance was positively correlated with ambulation (p = .466, p = .002). A significant negative correlation was found between low physical activity and ambulation (p= −.325, p = .041). No other significant associations were observed among the frailty components.
- Did you run any Sensitivity/Specificity tests using Fried phenotype as reference standard? Please run these tests and report
Thank you for the insightful suggestion. We understand the value of evaluating sensitivity and specificity using the Fried phenotype as a reference standard. However, the purpose of this study was not to validate an alternative frailty measure against Fried phenotype but rather to explore the differences between Fried phenotype and K-FRAIL scale. Furthermore, Fried phenotype have used as an assessment tool in previous studies (Fried et al., 2001; Fried et al., 2004; Kaya & Yavuz., 2022). We believe that running sensitivity and specificity analyses would not align with the goals or design of the current study. We hope this clarification is acceptable and we respectfully suggest that these analyses may be better suited for a future validation focused study.
Reference: 1. Fried, L.P., et al., Frailty in older adults: evidence for a phenotype. J Gerontol A Biol Sci Med Sci, 2001. 56(3): p. M146-56.
- Fried, L. P., Ferrucci, L., Darer, J., Williamson, J. D., & Anderson, G., Untangling the concepts of disability, frailty, and comorbidity: Implications for improved targeting and care. The Journals of Gerontology: Series A, Biological Sciences and Medical Sciences, 59(3), 255–263.
- Kaya, D., & Yavuz, B. B., Validity and reliability of Fried frailty phenotype in Turkish population. Aging Clinical and Experimental Research, 34(10), 2591–2598.
7. Report the test you used for frailty prevalence comparisons? McNemar-Bowker test for differences in classification proportions?
We appreciate your insights on the statistics. We conducted the McNemar test to compare the prevalence of frailty. However, there were no participants classified as frail according to the K-FRAIL scale, the expected frequency was less than 5, and the statistic test could not be calculated (expected frequency = .68). When comparing the frailty prevalence (%) of the two assessment tools, the frailty phenotype identified 7.5% as frail, while the K-FRAIL scale identified none (0%). We suggest that the two assessment tools apply different evaluation criteria and reflect differences in their ability to identify individuals as frail.
- Disclose handling of missing data and inter-rater reliability
Thank you for your comment. In our study, there were no missing or incomplete data and all participants completed every assessment item. To ensure consistency and reduce potential variability, a single researcher evaluated all evaluations of the two assessment tools.

Round 2
Reviewer 1 Report
Comments and Suggestions for Authors
I thank the Authors for addressing all of my comments.
Author Response
Thank you once again for your valuable comments and kind support throughout the review process.
Reviewer 2 Report
Comments and Suggestions for Authors
MINOR REVISIONS
- Please convert all tables into word format for better formatting. Currently, the tables are embedded as images and look unclear.
- Line 71: "2.1. The Participants" should be revised to "2.1. Participant eligibility criteria"
- Line 176: "3.1. The Characteristics of the Participants" should be revised to "3.1. Participant characteristics"
Author Response
Thank you once again for your valuable comments and kind support throughout the review process.
We have converted the table into Word format and made the necessary revisions which have been highlighted in the manuscript.
